# Tumor Classification Should Be Based on Biology and Not Consensus: Re-Defining Tumors Based on Biology May Accelerate Progress, An Experience of Gastric Cancer

**DOI:** 10.3390/cancers13133159

**Published:** 2021-06-24

**Authors:** Helge Waldum

**Affiliations:** Department of Clinical and Molecular Medicine, Faculty of Medicine and Health Sciences, Norwegian University of Science and Technology, 7030 Trondheim, Norway; helge.waldum@ntnu.no

**Keywords:** cell of origin, cancer, growth regulation, hormones, tumor classification

## Abstract

**Simple Summary:**

Rational treatment of diseases including cancers depends on knowledge of their cause as well as their development. The present review is based upon more than 40 years’ work in clinical gastroenterology, gastric physiology, and pathology. The central role of hormones as well as local endocrine cells in cancer development has become apparent. Moreover, the classification of tumors should focus not only on the organ of origin but also on the cell of origin. All cells with the ability to divide may give rise to tumors. Based upon knowledge of the growth regulation of the cell of origin, prophylaxis and treatment may be tailored. Presently, there is hope for individual treatment of cancer patients based upon genetic analyses of tumors. However, with correct identification of the cell of origin, this may not be necessary.

**Abstract:**

Malignant tumors are a consequence of genetic changes mainly occurring during cell division, sometimes with a congenital component. Therefore, accelerated cell divisions will necessarily predispose individuals, whether due to conditions of chronic cell destruction or hormonal overstimulation. It has been postulated that two genetic hits are necessary for the development of malignancy (Knudson). The correct view is probably that the number of genetic changes needed depends on the role the mutated genes have in proliferation and growth control. Hormones should accordingly be regarded as complete carcinogens. In this review based upon experience of gastric cancer where gastrin is central in the pathogenesis, it is argued that oxyntic atrophy—and not metaplasia as postulated by Correa—is the central precancer change in gastric mucosa. Moreover, the target cell of gastrin, the enterochromaffin-like (ECL) cell, is central in gastric carcinogenesis and most probably the cell of origin of gastric carcinomas of the diffuse type according to Lauren (a classification probable in accordance with biology). The distinction between adenocarcinomas and neuroendocrine carcinomas based upon a certain percentage of cancer cells with neuroendocrine differentiation is questioned. To make progress in the treatment of cancer, a correct classification system and knowledge of the pathogenesis are necessary.

## 1. Introduction

Cancers or malignant neoplasms are diseases where cells no longer respect their border for normal distribution but grow into neighboring structures (invasion) and/or invade other organs via transport in the blood or lymphatics or via natural cavities (metastasis). These properties are due to genetic changes most often representing the loss of some control mechanism. However, even if it is undisputed that there are genetic changes in cancer cells, only a minor proportion of cancers occur due to congenital genetic disorders [1,2]. The genetic changes leading to cancer most often develop with time in proliferating cells. Each cell division represents a small risk of genetic error (mutation) with an estimated frequency of 3.0 × 10^−8^ mutations/nucleotide/generation [3]. Mutations are the main mechanism for evolution as well as cancer development. The mutation rate in man is very low, partly due to repair mechanisms where DNA mismatch bases are excluded and replaced by the correct ones [4]. Nevertheless, some mutations escape through the control system and are incorporated into the cell. Most of these mutations have no functional importance or at least do not predispose the individual to cancer development [5]. However, with time, mutations are accumulated, and some may, alone or in combination with other mutations, be crucial in cellular growth regulation and transform the cell into malignancy. According to the two-hit theory of carcinogenesis, at least two mutations are required for a cell to become malignant [6]. Since each cell division implies a certain but minor risk of mutation, it is strange that stimulators of proliferation like hormones are not accepted as complete carcinogens. In this short review, I will also challenge the theory that stem cells are the major cells of origin of cancers, arguing that all cells able to divide may develop into a tumor. Finally, I will focus on the role of gastrin and its target cell, the enterochromaffin-like (ECL) cell, in gastric carcinogenesis, and thereby demonstrate the importance of improved classification of tumors to improve understanding and treatment. The dominating role of gastrin in gastric carcinogenesis [7] also shows the important role of hormones in carcinogenesis. This paper is based upon life-long experience and a thorough search in PubMed where relevant articles, whether supporting or opposing my view, were included.

### 1.1. Tumorigenesis/Carcinogenesis

It is fascinating how the number/mass of the different cell types is controlled, leading to constant values in steady-state conditions. This observation led Bullough, in 1965, to postulate the presence of a specific substance (chalone) released from each cell type with an inhibitory effect on the proliferation of the same cell type [8]. This concept seemed important but gradually lost interest since no chalone was purified and identified. However, in 1997, a TGF-like molecule, Gdf8 (also named myostatin), occurring in skeletal muscle, was identified, and knock-out mice were found to develop a dramatic increase in muscle mass [9]. This substance thus had properties compatible with the old chalone concept, and since, other chalone candidates have been described for other tissues/cells [10]. Although the lack of a chalone could be postulated to predispose an individual to tumorigenesis, presently, no such case has been described. Hormones, on the other hand, often stimulate the proliferation of certain cell types, leading to an increased mass of that cell type. Even during hormonal stimulation, a new steady state is reached, suggesting that inhibitory mechanisms are still operating, although with a new set point [11]. Although the density of a target cell exposed to continuous hormonal overstimulation reaches a maximum, some target cells show increased proliferation, leading to local accumulation of cells in clusters, representing dysplasia and the start of neoplasia [12]. The target cells growing in clusters obviously must have undergone some form of change in the regulation of proliferation, which is transfected to the daughter cells as a genetic change. By normalization of the hormone level, such histological nests or even macroscopic tumors at early phases may apparently disappear [13]. However, it is plausible that some mutated cells persist although they are not detectable at the actual time. Thus, treatment of gastrin-induced ECL cell neuroendocrine tumors (NETs) with a long-acting somatostatin analog made them disappear, but they reappeared relatively shortly after stopping treatment [14].

The best model for studying this and showing that hormonal overstimulation alone is sufficient to induce neoplasia is related to the role of gastrin in ECL cell hyperplasia, dysplasia, and neoplasia [15,16]. Thus, hypergastrinemia in man due to congenital anacidity [17], gastrinoma (whether part of multiple endocrine neoplasia type I (MEN I) [18] or sporadic [19,20]), autoimmune gastritis [21,22], long-term profound acid inhibition [13,23,24], Helicobacter pylori infection [25,26], or surgery leaving antral mucosa unexposed to acid [27] predisposes the individual to ECL cell hyperplasia and neoplasia. Similarly, hypergastrinemia causes gastric neoplasia of different degrees of malignancy in rodents [15,16,28,29,30]. Hypergastrinemia is the only common factor in all these conditions, clearly showing that hypergastrinemia alone is sufficient to induce gastric neoplasia of all degrees of malignancy; thus, gastrin is a complete carcinogen. Moreover, gastrin-driven gastric carcinogenesis shows that neoplasia including cancer may develop from differentiated cells with the ability to divide, as exemplified by the ECL cell [17,31]. Furthermore, the ECL cell-derived tumors demonstrate hormonal carcinogenesis, as was realized already in the mid-1980s [32]. In the latter study, the authors could not show any differences between the tumor cells in gastric NETs due to gastrinoma or autoimmune gastritis [32], which casts doubt on the wisdom of classifying them into two different subtypes, as later proposed [33]. Moreover, since all conditions with long-term hypergastrinemia predispose individuals to gastric NETs, there is no need to claim that inflammation plays an additional role to gastrin in the pathogenesis of gastric NETs in patients with autoimmune gastritis. The role of gastrin in gastric disorders is also reflected in its inclusion as one of the parameters in a commercial test [34]. Individuals with elevated gastrin have been shown to have an increased risk of developing gastric cancer [35,36], and some studies have reported increased gastrin in patients with gastric cancer [37,38,39]. However, it is true that gastrin apparently may be normal in many patients with gastric carcinoma, which could suggest that gastrin nevertheless is not central in gastric carcinogenesis. But the upper normal level for gastrin hitherto has been too high (since persons with H. pylori gastritis were included in the normal material when gastrin immune assays were established [40] (J.F. Rehfeld personal communication)). Therefore, gastrin has been elevated in many individuals characterized as normogastrinemic. Furthermore, fasting gastrin value underestimates 24-h gastrin [41], and gastrin is very potent, showing a steep trophic-effect curve [42] without a threshold [43].

The theory of Knudson that it is necessary with two mutations for the development of a cancer cell [6] probably only applies to tumor suppressor genes that are recessively inherited, needing loss of function of both alleles. It is conceivable that an activating mutation in a single allele of a proto-oncogen with a central position in carcinogenesis could be sufficient to induce a malignant tumor. In the stomach, type-III ECL cell-derived NETs [33] develop without a background of hypergastrinemia and may be postulated to be due to such a central mutation. However, new mutations, many with minor impacts on tumor behavior, will of course, occur after the initial central mutation [44]. On the other hand, during hormonal carcinogenesis due to increased proliferation, mutations occur and are progressively accumulated. Some of the mutations may have a slight tumorigenic effect and their additive effect may give properties to the cell in the direction of tumor development. The gradual transition from a rather benign to a highly malignant tumor is well-known for ECL-cell NETs [31,45]. Generally, NETs have been shown to have increased proliferation with time [46] and develop into more malignancy [47]. Surprisingly few studies on the histological change in tumors during this process exist. In a study from Italy, entero-pancreatic neuroendocrine neoplasms (NENs) new biopsies were re-assessed at the time of clinically progressive disease, from 3 to 128 months after the initial sampling. They found that grading and Ki67 increased in those with pancreatic NET, but not in those located in the small intestine [48]. However, different responses to treatment may have affected those findings. Nevertheless, NETs, like other tumors, can develop into more malignant tumors. To my knowledge, peptide hormones have been shown to have a direct tumorigenic effect only on neuroendocrine/endocrine cells. Lipid-soluble hormones like estrogens play an important role in carcinogenesis in reproductive organs, causing cancer of the vagina at a young age in girls born to mothers treated with estrogen during pregnancy [49].

There is an accepted fact that NETs phenotypically are very similar to the normal cell of origin. Nevertheless, they can invade and metastasize at a stage with slow proliferation. The slow proliferation of normal neuroendocrine cells led to the suggestion that they did not divide in man [50], and neuroendocrine cells in adenocarcinomas were also described without this ability [51]. The latter finding was based upon the lack of proliferation of chromogranin A-positive cells. It is possible chromogranin A was not expressed in the phase of cell division. The question of proliferation of neuroendocrine cells was particularly focused on the gastric ECL cell in man, which certainly does proliferate—as shown by specific development of hyperplasia upon gastrin stimulation [23]—and thus is similar to rodent ECL cells [52]. In general, it should be recalled that men and rodents are more than 90% genetically identical, and therefore, a claim that man is so special that findings in animals must be shown in man before being accepted to have clinical relevance, is false. Such a view will certainly retard the recognizing of drug side effects with a long latency [53].

### 1.2. Classification of Tumors and Cell of Origin

Pathological examination of tumors is the basis of the classification and treatment of tumors. Rudolf Virchow (1821–1902) started the microscopic examination of tissue, and since his time, new techniques have continuously been developed, improving classification. Nevertheless, the initial methods—with fixation of tissue followed by cutting into thin slices put on glass plates, subsequently exposed to chemicals with specific affinities to molecules (structures) before microscopic examination—are still the basis of histochemistry in routine use. However, there are many strange and obviously unfortunate conventions in the classification system. For instance, gastric carcinomas are divided into proximal and distal locations, where the proximal ones are thought to originate from the tiny cardiac portion and distal ones are lumped together, originating from two different mucosae, the oxyntic and antral mucosa. The classification should obviously be cardiac, oxyntic, and antral mucosa, which would reflect the tissue of origin. The central role of the ECL cell, which is found only in the oxyntic mucosa, is an argument in favor of such a change [36,54].

In this context, I will also mention the hypothesis of Correa that gastric carcinomas develop from intestinal metaplasia through a sequence of gastritis, atrophy, and intestinal metaplasia [55]. For Helicobacter pylori gastritis, it has been clearly shown that oxyntic atrophy is the crucial step in gastric carcinogenesis [26], and that in persons with oxyntic atrophy, whether due to previous Helicobacter pylori infection [56,57] or autoimmune gastritis [58,59], the risk of neoplasia continues even after Helicobacter pylori eradication in those with H. pylori-induced atrophy. Intestinal metaplasia may just be a marker of cancer risk and not a direct precursor [60]. The oxyntic atrophy causes hypoacidity, leading to hypergastrinemia, which is central in the carcinogenesis induced by Helicobacter pylori [7] as well as autoimmune gastritis [61], and also explains the risk of long-term profound acid inhibition [62].

### 1.3. The Distinction between Adenocarcinomas and Neuroendocrine Carcinomas

Now, I will turn to the question of why the percentage of NE cells in carcinoma is of importance in distinguishing between adenocarcinoma and neuroendocrine carcinoma (NEC). In the current WHO classification, tumors with more than 30% of the tumor cells expressing neuroendocrine markers are classified as NECs or mixed adeno-neuroendocrine neoplasms (MiNEN), and those with less than this percentage are classified as adenocarcinomas [63]. Such a classification will depend on the sensitivity of the methods used to detect the markers. From a biological point of view, this seems very peculiar. Moreover, there is a distinction between NETs and NECs, although it is well-known that tumors belonging to the NET group can grow invasively and metastasize. Until recently, NETs were called carcinoids because of their similarities to cancers. In many ways, NETs are malignant tumors, although with a better prognosis than many cancers, and the logical name would be neuroendocrine carcinoma of low grade. Jun Soga, one of the pioneers within neuroendocrine pathology, also suggested that carcinoma should be part of the name [64]. The classification problems related to neuroendocrine neoplasia are connected to insufficient knowledge of the cell of origin. For instance, postulating that the ECL cell in the stomach gives rise to NETs but not NECs is strange taking into consideration that most benign tumors generally tend to develop into more malignant ones with time. Moreover, in a Spanish family with a lack of gastric acidity from birth, ECL cell tumors occur in the third decade, and more malignant tumors classified as adenocarcinomas in individuals’ at thirties [17]. The adenocarcinoma was later reclassified as NEC [65]. Similarly, we have shown how a NET from the ECL cell in a patient with pernicious anemia progressed to become a very malignant tumor [31], and that the carcinomas in patients with long-term hypergastrinemia expressed neuroendocrine markers when studied with immunohistochemistry with improved sensitivity [45]. Therefore, the hypothesis by Solcia et al. [66] that the gastric ECL cell does not progress to malignant tumors beyond NETs is not true. In fact, in the late 1970s, Solcia et al. presented the opposite view [67]. We have previously advocated for a new classification of cancers, and neuroendocrine tumors, in particular [68].

The classification of gastric carcinomas into adenocarcinomas of the intestinal type with glandular growth pattern and diffuse without such growth [69] seems to represent important biological differences since these two types do not transform into one another. The inclusion of the diffuse type in adenocarcinomas was based upon PAS/Alcian-blue positivity, regarded as specific markers of mucin, a substance expressed in exocrine cells. However, these staining methods are not specific to mucins. PAS, for instance, binds to glycoprotein/peptides in general [70]. The question of classification of signet-ring cell carcinomas, a subgroup of diffuse-type carcinomas, among adenocarcinomas has been problematic. Thus, we found that PAS-positive cancer cells expressed neuroendocrine markers, suggesting that the tumors were NECs [71]. In another study, the neuroendocrine marker chromogranin A was expressed in an important portion of signet-ring cell carcinomas, and patients had better prognoses if chromogranin A was expressed in the cancer cells [72], indicating that this expression was lost during the process of malignancy. Moreover, signet-ring cells need not express PAS/Alcian-blue positivity but may be strongly positive for neuroendocrine markers [73,74]. Interestingly, signet-ring cell carcinoma secondary to hypergastrinemia has been described [75], further strengthening the assumption that signet-ring cell carcinomas are NECs and often of ECL cell origin. In general, gastric carcinomas of the diffuse type contain tumor cells expressing neuroendocrine markers significantly more often than those of the intestinal type [76]. It has, however, to be admitted that there is a group of gastric carcinomas that cannot be classified as either intestinal or diffuse type, showing traits of both types. This may be explained by the ability of NE cell tumors to display a glandular structure [77]. Finally, E-cadherin loss was reported to give more lymph-node metastasis in neuroendocrine gastric carcinomas [78]. We have previously reported that normal gastric ECL cells do not express E-cadherin [79].

The reason for classifying gastric carcinomas of the diffuse type among adenocarcinomas was the positivity for PAS/Alcian-blue of that time, thought to be rather specific for mucin, although that is not the case [70]. Furthermore, the homogenous material in gastric signet-ring cells is negative for mucins by immunohistochemistry as well as in-situ hybridization [80]. Quite recently, it was reported that diffuse gastric carcinomas and signet-ring cell carcinomas were clonally identical and that they could transform into each other upon changes in the tumor environment [81]. Accordingly, gastric carcinomas of the diffuse type do not express any undisputed exocrine marker, whereas markers of neuroendocrine cells are found. Logically, they should, therefore, be classified as NECs [54,82].

### 1.4. The ECL Cell

The dominating neuroendocrine cell in the oxyntic mucosa is the ECL cell, which is also the only cell with an undisputed gastrin receptor [83]. However, gastrin has—besides a specific effect on the ECL cell—also a less pronounced, general trophic effect on the oxyntic mucosa [84,85,86]. Thus, there is no specific trophic effect on parietal cells. Whether there is a gastrin receptor on the stem cell, or the general trophic effect is transmitted indirectly by the release of a signal substance from stimulated ECL cells, is not yet clear. Among the signal substances released from the ECL cell with an important proliferative effect are Reg proteins [87]. Interestingly, Reg proteins have been reported to increase proliferation along parietal and chief cell lines, but not ECL cells [88], which is in agreement with gastrin having a general trophic effect via stimulation of the gastrin receptor on the ECL cell, leading to Reg proteins’ release, stimulating proliferation of the stem cell. Such a mechanism may also explain why hypergastrinemia is central in gastric carcinogenesis of the diffuse type, originating from the ECL cell and in carcinomas of the intestinal type, via Reg proteins and the stem cell [54].

Basic fibroblast growth factor (bFGF) is another signal substance released by the ECL cell that has an effect on neighboring cells [89]. Production of bFGF by gastric NETs has for a long time been known [89], and there is a report of a gastric scirrhous carcinoma producing bFGF, thus inducing aggressive fibrosis around metastases [90]. In scirrhous carcinoma, bFGF was reported to show increased expression compared with other types of gastric cancer as well as normal mucosa [91]. Fibrosis is a typical trait of gastric carcinomas of the diffuse type, and bFGF may be an important contributor to this tendency.

### 1.5. Gastric Physiology with Relevance to Gastric Cancer

In the above, I have made arguments in favor of the biological relevance of Lauren’s classification [69] and the central role of gastrin in gastric carcinogenesis. Gastrin, a peptide hormone, has a direct effect only on cells with a gastrin receptor, and thus only the ECL cell [83]. There has been a long, and partly emotional, debate about a direct effect of gastrin on the parietal cell. There was a legendary dispute between Code [92] and Grossman [93] on the interaction between the major gastric acid secretagogues. Swedish authors found that gastrin, in contrast to histamine and cholinergic agents, did not stimulate acid secretion in isolated oxyntic glands and isolated parietal cells [94], thus supporting the view of Code that gastrin had an indirect effect. However, Soll, studying isolated canine cells, claimed a faint effect by gastrin, although the statistical evaluation was somewhat dubious for excluding non-successful studies [95]. In a completely isolated rat stomach, we found that maximal gastrin-stimulated acid secretion was inferior to that by histamine, and that gastrin—in contrast to a cholinergic substance—did not augment maximal histamine-stimulated acid secretion [96]. When we determined histamine by a radio-immune assay [97]), we could show that the histamine release was sufficient to explain the stimulation of acid secretion by gastrin [98]. However, Kopin et al. cloned the gastrin receptor from oxyntic mucosal cells enriched, but not completely pure, in parietal cells [99]. Using a gastrin analog in a concentration in the physiological range, we demonstrated that the ECL cell and not the parietal cell expressed the gastrin receptor [83]. Quite recently, there was a study showing the effect of gastrin receptor expression on ECL cells but also on pre-parietal cells, though not on chief cell precursors [100]. This is very strange since gastrin has neither a specific trophic effect [85,86] nor a functional effect on the parietal cell [94,96]. Therefore, I will conclude that there is no gastrin receptor on the parietal cells or its specific precursors. However, gastrin indirectly stimulates the proliferation of the parietal cell as well as the other cell types in the oxyntic mucosa except for the ECL cell. The general trophic effect of gastrin on non-ECL cells in the oxyntic mucosa is presumably due to indirect stimulation of the common stem cell. Nevertheless, the location of the gastrin receptor has remained controversial. Although stimulation with gastrin induces hyperplasia, and local accumulations of ECL cells [12] and mature ECL cells have been shown to proliferate [101], Sheng and co-workers claimed that the increase in ECL cell density due to hypergastrinemia was mainly due to the proliferation of precursors (stem cells) in the neck area of the oxyntic glands, with some specificity for parietal cell precursors [100]. As there is no indication of gastrin having a special function on the parietal cell, this would indicate the preservation of a functionless mechanism. Moreover, Sheng et al. indicated that there was the proliferation of ECL cells both by differentiation of probable stem cells as well as mature ECL cells, both carrying gastrin receptors [100]. Taking into consideration that ECL cells release compounds like Reg proteins stimulating proliferation of the stem cell [87,88], a separate proliferative effect via a gastrin receptor, both on stem cells and mature ECL cells, seems rather complicated. It has long been known that gastrin has a stronger proliferative effect on the ECL cell compared with the other oxyntic mucosal cells including the parietal cells [85,86]. It may accordingly be concluded that mature ECL cells have the gastrin receptor and that mature ECL cells do proliferate themselves; the general trophic effect on the oxyntic mucosa by gastrin may be indirectly mediated by signal substances released from stimulated ECL cells or possibly by a presumptive gastrin receptor on the stem cell. The probable mechanism for gastrin-induced gastric cancer is depicted in Figure 1.

In this review, I have focused on gastric cancer, challenging many hypotheses and paradigms (Table 1). However, as written before, many of the arguments also have relevance for tumors in other organs [102], and we made a paper 13 years ago, together with Jun Soga, covering some of the elements in the present paper [68]. The clinical importance of correct classification of the cell of origin (biologically based classification) is best exemplified for gastric cancer where the growth regulation of the normal ECL cell explains gastric carcinogenesis. Knowledge of the mechanisms gives indications for age for eradicating H. pylori (before development of oxyntic atrophy) and when to take a gastrin antagonist like netazepide [103,104] in prophylaxis and the treatment of gastric malignancies of a moderate grade (NETs) [104], and possibly of gastric carcinomas as well since they often express the gastrin receptor [105]. Moreover, in the treatment of gastroesophageal reflux disease, the reduction of gastric acidity should be as low as possible to reduce hypergastrinemia. Similarly, we found, in renal carcinomas of a clear cell type, that virtually all cancers expressed erythropoietin [106], which indicates that the erythropoietin cell is the cell of origin and which could also explain the role of hypoxia and hypoxia-inducible transcription factors (HIF) in clear-cell renal cancer [107]. This concept is about to have a therapeutic implication [108]. Above, I have given two examples of important cancers—gastric cancer and clear-cell renal cancer—where knowledge of the cell of origin gives important indications for treatment; as such, there is every reason to believe that identification of the cell of origin will have a similar effect for cancers in other organs.

Some years ago, a classification of gastric carcinomas based upon genetic changes was reported [109]. Such a classification may be useful for the determination of treatment possibilities but does not tell us much about the cell of origin or pathogenesis since most of the genetic changes in a cancer cell have occurred, by chance, secondary to the influence of a carcinogen or as a consequence of a normal cell division.

It may be argued that a change in classification from the organ to the cell of origin may require a multidisciplinary approach with the involvement of clinicians including oncologists. However, the experience is that colleagues easily adapt to new knowledge that makes practice more rational and improves treatment.

## 2. Conclusions

Hypotheses and paradigms are formulated at a certain stage of knowledge and may be useful for a period. They should not be kept when new findings are incompatible with them. In other words, hypotheses and paradigms should not be treated as holy. Knowledge of the cell of origin of cancers gives indications of the tumor pathogenesis based upon the growth regulation of the cell, and similarly, possible treatment in stages of the tumor where actual receptors are still expressed. Thus, gastrin antagonists have not been developed for clinical use because the role of gastrin in gastric carcinogenesis has been denied. Netazepide, a specific gastrin antagonist with few if any side effects and proven efficacy in the treatment of gastric NETs, has not been tried in the treatment of gastric cancers although gastrin receptors are frequently expressed. The detection of the erythropoietin cell as the possible cell of origin of clear-cell renal cancer may be related to the role of HIF in renal carcinogenesis, and may hopefully result in new treatment options for this cancer. It is likely that tumor classification using available markers with the highest sensitivity will also disclose an unexpected cell of origin for cancers at other locations.

## Figures and Tables

**Figure 1 cancers-13-03159-f001:**
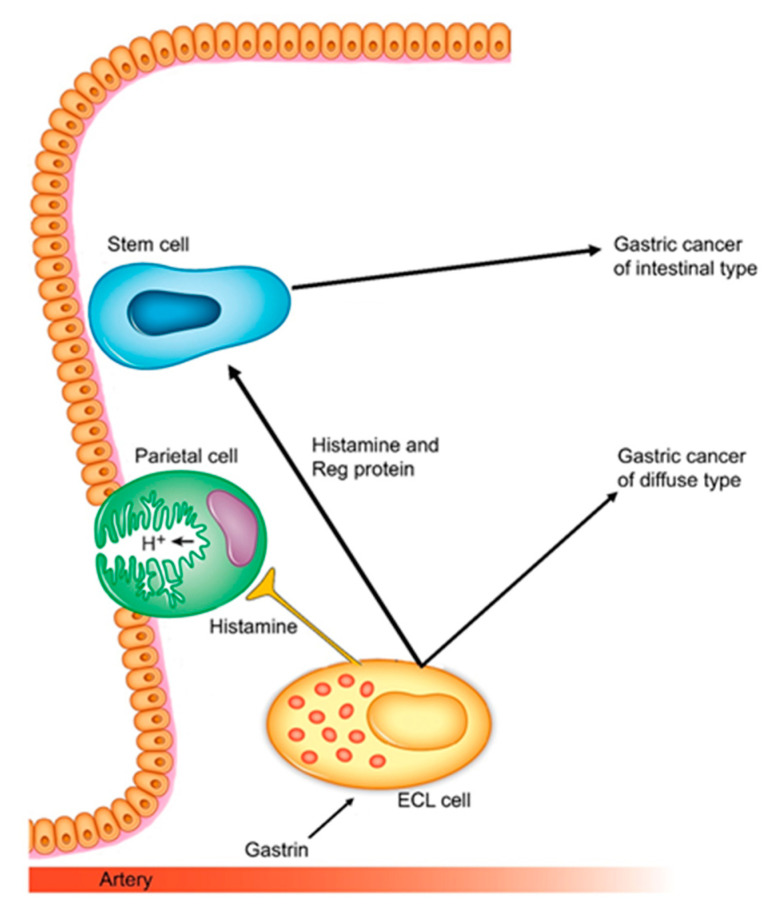
Gastrin and the ECL cell are central in gastric physiology and carcinogenesis [54].

**Table 1 cancers-13-03159-t001:** Challenging old hypotheses and paradigms.

• Gastric carcinomas should be located to the three different mucosae (cardiac, oxyntic, and antral) and not as being proximal (cardiac) and distal (oxyntic and antral).
• The central role of metaplasia in gastric carcinogenesis according to Correa should be replaced by oxyntic atrophy. Metaplasia may just be a marker of advanced longstanding atrophy.
• Lauren’s classification represents biological differences with an intestinal type probably originating from the stem cell and a diffuse type from the ECL cell.
• The percentage of tumor cells expressing a marker depends on the sensitivity of the method used. Setting a certain figure to differentiate between, for instance, adenocarcinomas and neuroendocrine carcinomas, is not logical.
• PAS/Alcian-blue positivity is not specific for mucin and thus does not imply exocrine origin.
• Gastrin is central in gastric carcinogenesis whether due to Helicobacter pylori or autoimmune atrophic gastritis as well as profound acid inhibition. Gastrin is a complete carcinogen for the oxyntic mucosa.
• The wisdom of subclassification of gastric NETs due to hypergastrinemia into two types is challenged.
• The gastrin receptor is localized on the ECL cell and possibly on the stem cell.
• The two-hit theory for carcinogenesis by Knudson is useful for understanding some congenital cancers, but in general, cancer may develop due to a single mutation affecting a central function of the cell, although most cancers occur due to multiple mutations, which together move the cell into malignancy.
• Hormones are complete carcinogens by their stimulation of proliferation causing an increased rate of mutations.

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
