# Peer review of "Tumor Classification Should Be Based on Biology and Not Consensus: Re-Defining Tumors Based on Biology May Accelerate Progress, An Experience of Gastric Cancer"

_cancers, 2021, doi:10.3390/cancers13133159_

Round 1

Reviewer 1 Report

GENERAL COMMENTS

The Author is proposing a new classification of gastric cancer, based on cell of origin rather than on organ of origin. He maintains that: 1) Oxyntic atrophy rather than intestinal metaplasia in the most important precancer change in gastric mucosa. 2) ECL cells play a key role in gastric carcinogenesis, as they directly give rise to diffuse cancer and indirectly stimulates stem cells to originate intestinal cancers. 3) Percentages of neuroendocrine cells is not a good criterion to distinguish between adenocarcinomas and neuroendocrine tumours.

The review is convincing, and based on a large review of the scientific literature. The Author also gives an historical perspective. I had not heard about chalones, since my medical studies.

MAJOR PROBLEM

1.The present review is restricted to the histopathology of gastric carcinogenesis. However, the Author tries to give a more general meaning, starting from the title “Tumour classification should be based on biology and not consensus”. In the Simple Summary: “Presently there is hope for individual treatment of cancer patients based upon genetic analyses of the tumours. However, by a correct identification of the cell of origin this may not be necessary”. In the Abstract: “Hormones should accordingly be regarded as complete carcinogenesis”.

Changing the paradigm of gastric cancer would require a multidisciplinary team with a multidisciplinary approach, including surgery, medical oncology, pathology, genomics, while the present review is mainly based on pathology. For instance, a more comprehensive approach would require to take into account the evolution of TNM classifications over time. The AJCC-UICC pN classification of gastric cancer was based on the distance of metastatic nodes from the primary tumour in 1987, and on the number of the involved nodes in 1997. In 2010 cancers originating within 5 centimetres from the cardia were included among oesophageal tumours. Moreover, the scientific community is currently debating whether and how to move from a cancer classification based on the organ affected and on histopathology to a new one, based on genetic disorders, and this change of paradigm is occurring also for gastric cancer [The Cancer Genome Atlas Research Network, 2014].

In my opinion the Author should concentrate on histopathology of gastric carcinogenesis, and should be careful in proposing a new comprehensive paradigm.

SPECIFIC PROBLEMS

2.The role of gastrin in gastric carcinogenesis has been acknowledged also in the diagnostic field. Gastrin-17b has been included in the set of biomarkers, named Gastropanel® (Copyright© 2020, International Institute of Anticancer Research. Delinasios GJ), which Pepsinogen I, Pepsinogen II, gastrin-17b and IgG to H. Pylori.

3.I question the sentence “Hormones should accordingly be regarded as complete carcinogenesis”. This seems true for gastrin and gastric cancer, it’s probably true for insulin and colorectal cancer. However, lung cancer is mainly due to cigarette smoking, alcohol has a role in promoting SCC of the proximal oesophagus, obesity and GERD are linked to adenocarcinoma of the distal oesophagus. Diet has an important effect on gastric carcinogenesis: meat consumption increases the risk of gastric cancer, while citrus and polyphenols are protective factors.

4.As regards site of origin, the Author states that “the classification should obviously be cardiac, oxyntic, and antral mucosa which would reflect the tissue of origin”. However, an anatomical classification in cardia, upper, middle and lower third is very important to infer lymphatic spread and decide the surgical approach accordingly.

REFERENCES

The Cancer Genome Atlas Research Network., Analysis Working Group: Dana-Farber Cancer Institute., Bass, A. et al. Comprehensive molecular characterization of gastric adenocarcinoma. Nature 513, 202–209 (2014). https://doi.org/10.1038/nature13480

Author Response

Thank you for important suggestions.

General comments.. (1). Oxyntic atrophy versus intestinal metaplasia in gastric carcinogenesis. Lines 173-183. (2) The role of the ECL cell in gastric carcinogenesis cannot be separated from gastric physiology especially the location of the gastrin receptor. Lines 248-319. (3). Percentage of neuroendocrine cells in the classification.. Lines 184-190.

  1. Apparently normal gastrin values in patients with gastric cancer. This is  a very important question. However, the upper level for normal gastrin is presently too high since persons with H. pylori gastritis were included in the normal material at the time gastrin immunoassays were established. The will probably result in a paper by Professor Jens F. Rehfeld on this topic.. This subject is commented on in : Lines 112-119.
  2. Clinical and therapeutic implications: Lines 323-339 

Reviewer 2 Report

It was my honor to review the article entitled “Tumour classification should be based on biology and not consensus. Keeping old paradigms retards progress. Experience based upon gastric cancer”.

This is a very interesting review conducted by an expert in the field.

The paper summarizes the findings of research on gastric carcinogenesis, on tumor classification and on the role of gastrin as a stimulator of ECL cells proliferation. Some “dogma” such as Knudson and Correa hypothesis were addressed giving a significant contribution in the field.

I have only two comments:

1) According to the author, hypergastrinemia following oxyntic atrophy, is central in the carcinogenesis induced by HP or autoimmune gastritis. Hypergastrinemia alone is sufficient to induce gastric neoplasia and therfore it can be considered a complete carcinogen. Most gastric NETs are composed of histamine-producing ECL cells. However, hypergastrinemia was also associated with an increased risk of gastric adenocarcinoma (ref. 36). How can normal serum gastrin levels be interpreted in patients with gastric cancer?

2) I think it is interesting to add a brief comment on the clinical and therapeutic implications in the light of these findings.

Author Response

Thank you for useful comments.

Major problem 1. Classification based upon genetic changes;: Lines 340-344. Problems for clinicians including oncologists to change classification: 345-348. Hormones as complete carcinogens. Since bad luck (chance) during cell division perhaps is the most common cause of cancer, every stimulation of cell division which typically hormones do, will increase the risk and therefore in a way be complete carcinogens. This is extended in the introduction and also mentioning estrogen: Lines 140-144.

2. Specific Problems. The inclusion of gastrin in a  test for H. pylori. Lines  108-110. Reference 34 is included.

3. Hormones although being central in the pathogenesis of cancer, are of course the only factors. Even for gastric cancer, Helicobacter pylori is the main cause although the direct carcinogenic effect is mediated by gastrin. Tobacco smoking certainly is  one of the most important causes of cancer. The mechanism for and the cell of origin for pulmonary cancer due to tobacco smoking are not known. However, also in pulmonary cancer NE cells are important (oat cell cancer etc.). I have not included any discussion about tobacco smoking and pulmonary cancer, but I did in the late nineties two years  studies where we exposed rats for nicotine or carbon monoxide by air  without finding any trophic or neoplastic effect.

4. I have not made any changes related to classification according to normal mucosa. Although there are, of course, differences  in sites of metastasis, the classification proposed  does not make exact location more difficult.

The reference from Nature 2014 is included.

Reviewer 3 Report

This review examines literature related to classification and pathogenesis of tumors with emphasis on classification based on the cell of origin to inform cancer treatment. In particular, the review focuses on the role of gastrin and its target cell, the ECL cell, in gastric carcinogenesis and subsequently aims to demonstrates the importance of an improved classification of tumours to improve understanding and treatment.

There are interesting aspects of this review, however, I have some concerns related to the review:

  1. My main concern is that the review needs to be less theoretical and include more references to support statements contained within the review, especially in the introduction (specific instances are listed below).

  1. There is no outline of how papers were selected for inclusion in the review. That is, were databases (eg Pubmed, Cochrane) systematically searched to identify articles included in the review, or were certain papers supporting the argument of the author selected? This needs to be specified.

  1. The review states the findings will help inform cancer treatment. Could the authors include information on the applicability of the review findings. That is, how exactly will review findings improve cancer treatment? Recommend adding discussion re. the clinical implications of the review otherwise it is more theoretical than of clinical relevance.

Specific comments are listed below:

Page 1, line 3, title: Suggest not using “retards progress” but that re-defining tumors based on biology may accelerate progress. Also in the manuscript, the author does not suggest how the tumor classification could be changed to be “based on biology”

Page 1, line 36: Suggest using terms such as invading other organs, rather than to “leap”

Page 1-22, lines 35-56: The introduction is very theoretical and only includes one reference. There needs to be several references to provide evidence for the mutation rate in man, the role of DNA mismatch repair genes in reducing mutations, etc etc.

The introduction is also does not contain any background on the focus of the review (the role of gastrin and its target cell, the ECL cell, in gastric carcinogenesis) and the relevance of looking into this.

Page 3, line 102-103: Are there references to support this statement “The theory of Knudson that it is necessary with two mutations for the development 102 of a cancer cell (1) is an oversimplification of carcinogenesis.”

Page 3, lines 109: Re. the statement “On the other hand, during hormonal carcinogenesis due to increased proliferation, mutations occur and are progressively accumulated,” are there other studies apart from NETs?

Page 3, lines 120: Suggest removing the last sentence.

Page 5, Line 244-301: This paragraph is too long, suggest breaking it into 2 or 3 separate paragraphs.

Page 8, line 312: Suggest the conclusion be more specific to review findings. That is, the aim of the review was to highlight the importance of an improved classification of tumours to improve understanding and treatment but this is not addressed. That is, how does the author suggest tumors be classified and how will that help cancer treatment (specifically for gastric carcinogenesis, the focus of the review)

Author Response

Thank you for valuable comments.

Comments and suggestions

1.) New references are included among them five in the introduction. Hopefully this makes the manuscript less theoretical and more practical.

2.) How the review was made: Lines 61-62.

3.)Clinical utility. A new paragraph is added : Lines 321-336.

Specific comments.

a. The title is changes, thank you. As I see it, changing the classification to cell of origin from organ of origin is an important step in direction of biology. The whole manuscript is biologically based, but particularly the paragraph about physiology.

b. I have changed the word, thank you.

c.. I have included references as demanded, and I feel that doing so has improved the introduction. Moreover, there  is an explanation why dealing with gastrin and the ECL cell. Lines 57-60.

d. Two-hit theory  by Knudson. I have changed that description and added : Lines 121-124 and also in the Table.

e. Direct tumourigenic affect of hormones besides NETs. Lines 140-144.

f. I have "smoothened" the sentence: Lines 139-140.

g. I mean that the physiology is an important aspect of biology. Therefore I have kept the content, but reduced the chapter from 67 to 45 lines, thus shortened it by one third.

h.  The conclusion has been extended and focusing on the clinical utility of a more biologically oriented tumour classification. Lines 356-367

Round 2

Reviewer 1 Report

The Author reinforced his paradigm that tumour classification should be based on cell of origin. Indeed he also mentioned clear cell renal cancer, where erythropoietin seems to have the same role in carcinogenesis as gastrin in gastric cancer [By the way, dramatic improvement in renal cancer survival have been recently achieved by biological treatments, including immune checkpoint inhibitors].

The Author included clinical implications of his new paradigm, as requested. The Author described the commercial diagnostic use of gastrin level, citing the gastropanel article (I have no commercial nor scientific conflict of interest !!). He also mentioned and briefly discussed the implications of the new genetic classification of gastric carcinomas and briefly acknowledged the need for a multidisciplinary approach (lines 341-342).

It was again a pleasure to review this article. I was not aware of reference 49; while attending the Specialization School in Health Statistics during the Nineties, I was thought about another article dealing with in utero carcinogenesis caused by estrogen treatment: Herbst AL, Ulfelder H, Poskanzer DC. Adenocarcinoma of the Vagina — Association of Maternal Stilbestrol Therapy with Tumor Appearance in Young Women. N Engl J Med 1971; 284:878-881.

I still disagree with some of the Author’s statements, maybe because I have a different perspective, I am not a Pathologist. For instance, I disagree with the sentence in the Abstract: “Hormones should accordingly be regarded as complete carcinogenesis”. In my opinion this sentence is too general, for instance hormone replacement therapy in postmenopausal treatment has been shown to increase the incidence of breast cancer and decrease the risk of colorectal cancer [Rossouw JE, Anderson GL, Prentice Rl, et al (2002) Risks and benefits of estrogen plus progestin in healthy postmenopausal women – Principal results from the Women’s Health Initiative randomized controlled trial. JAMA 288:321-333]  

Line 343: a minor typo: HoTable 1.